# Anatomic and hemodynamic characterization of vertebral artery duplication via color doppler ultrasonography

Xue Han◦, Wenbo Duan◦, Wanling Wen, Tian Lin, Dongxu Lu, Juan Du*, Li Liu◦*

Department of Neurology, The Ninth Medical Center of General Hospital of the Chinese People's Liberation Army, Chaoyang, Beijing, China

◦ These authors contributed equally to this work and share the first authorship.

* snkjyb40207@sina.cn (LL); juanzijuanzi@263.net (JD)

## Abstract

### Background and objectives

Vertebral artery (VA) duplication is an infrequent vascular variation with a dearth of information on hemodynamic parameters. In this study, color Doppler ultrasonography was utilized to evaluate the origins and hemodynamic features of VA duplication to offer a solid reference for clinicians who are endeavoring to gain insights into this particular form of vascular anomaly.

### Methods

A retrospective analysis was conducted on patients with VA duplication detected by color Doppler ultrasonography. The analysis focused on the origins, the transverse course, diameter and the peak systolic velocity (PSV) of each segment of the duplicated VA.

### Results

In the 36 branches in 18 subjecte identified as VA duplication, 27 branches originated from the subclavian artery and 9 branches originated from the aortic arch. The incidence of variation in the non-C6 entry of the medial branch was 94.4%. Significant differences (P < 0.01) were found in the diameter between the lateral and medial branches, as well as between the medial branch and the convergent VA. Moreover, significant differences in PSV were noted between the medial and lateral branches (P = 0.035) and between the medial branch and the convergent VA (P = 0.015).

### Conclusions

VA duplication is correlated with substantial alterations in both the origin and the entry level into the transverse foramen. In our case series of VA duplication, the diameter

**Data availability statement:** Data were confidential to the public due to requirements for confidentiality. Data would be available from the Ethics Committee of the Ninth Medical Center of General hospital of PLA (contact via 8613521526535) for researchers who meet the criteria for access to confidential data.

**Funding:** This study was supported by the Subject Assistance Program of the Ninth Medical Center of the General Hospital of the Chinese People's Liberation Army (NO.21XK0101) The fund receiver did not take part in the study and had no role in study design, data collection and analysis, decision to publish, or preparation of the manuscript.

**Competing interests:** The authors have declared that no competing interests exist.

and PSV of the medial branch is different from that of the convergent VA or the lateral branch, which might be useful for neuro-interventionalists and neck surgeons.

## Introduction

Anatomical variations of the vertebral artery (VA) encountered in clinical practice may include abnormal origin of the VA, atypical entry point of the cervical vertebra into the transverse foramen, VA hypoplasia, and VA duplication. Among these, VA duplication is the rarest condition. In cases of VA duplication, two vertebral arteries arise from the proximal artery, and eventually converge into a single ascending branch. The incidence of unilateral VA duplication is estimated to 0.295%~0.79% [1,2]. The advancements in head and neck vascular imaging like color ultrasonography, magnetic resonance angiography, and computed tomography angiography have led to an increasing recognition of this clinically rare variation [2–4].

VA duplication has been reported to be associated with cerebrovascular diseases, such as aneurysms, arteriovenous fistula formation, arteriovenous malformation, and dissection [5,6]. The knowledge of this variance is of vital importance for therapeutic neurointervention and neck or spinal surgeries. Although several cases of VA duplication have been reported previously, the information about the possible anatomical variations in these cases is insufficient, especially hemodynamic parameters of the affected arteries. Herein, we analyzed the origin, course, internal diameter, and hemodynamic characteristics of VA duplication in 18 patients who underwent evaluation in our hospital using color Doppler ultrasonography.

## Materials and methods

### Patients

All examination records in patients who underwent VA color Doppler ultrasonography at a tertiary stroke center from January 2019 to December 2022 were reviewed. The indications for VA ultrasound examination were patients with complains of dizziness or vertigo that in suspicion of VA compression syndrome and patients with evidence or risk of ischemic cerebral vascular disease including but not limited to acute ischemic stroke or transient ischemic attack. The Ethics Committee of the hospital approved this study (protocol LL-LCSY-2024-06). The data are anonymous, and the requirement for informed consent was therefore waived.

### Ultrasonography examination

All examinations were performed using an ESAOTE color ultrasound machine with a probe frequency of 4–8 MHz. The patients were placed in the supine position, with a pillow placed under their shoulders and their heads extended as much as possible to ensure full exposure of the neck. The head was turned away from the side of detection. Then, the instrument probe was placed at the posterior border of the sternocleidomastoid muscle for longitudinal scanning. After scanning the common carotid artery (CCA) and its branches, the probe was adjusted to display the VA transversely.

The origin, course, entry level of the transverse foramen, diameter of each segment, blood flow velocity, and spectral pattern of the VA in each case were recorded. All patients were examined by an attending physician with more than 10 years of experience in cervical artery ultrasound examination. The ultrasound images of VA variations were evaluated by color Doppler ultrasonography to detect unilateral and duplicated VA origins.

### Data collection

The demographic and clinical data analyzed in this study included patient gender, age and comorbidities. Ultrasound parameters included the origin of the duplicated VA, and contralateral VA, the diameter of the duplicated trunk of the VA, the confluent VA and the contralateral VA in the intervertebral space segment. In addition, the peak systolic velocity (PSV) of the VA in the double-trunk VA and the corresponding measurements in the convergent intervertebral segment and the contralateral intervertebral segment were also analyzed. Standardized device parameters were ensured and preset optimizations were applied during PSV measurement: A unified preset condition for the VA was used. The Doppler gain was adjusted to clearly display the spectral waveform outline while avoiding background "snowy" noise. The wall filter was adjusted to retain true low-velocity flow signals while adequately filtering low-frequency signals caused by vessel wall motion. Doppler angle correction was consistently performed, with the correction line aligned parallel to the vessel wall. For the V2 segment of the VA, the straight portion between transverse foramina was generally used as a reference. The Doppler angle θ was maintained less than 60 degree. The sample volume size was set to cover most of the vessel lumen without extending beyond the vessel wall (typically 1.5–2.0 mm). It was positioned in the center of the vessel, away from near-wall regions, to obtain core flow velocities representative of laminar flow.

The medial branch of the duplicated VA was defined as the branch closer to the CCA, and the lateral branch was defined as the branch distal from the CCA.

### Statistical analysis

The results are presented as mean (SD) for continuous variables or number (%) for categorical variables. And they were compared using t – tests or chi – square tests as appropriate. The mean diameter and PSV of medial branches, lateral branches, confluent branches and the contralateral VA were compared between one another group.

## Results

### Patient characteristics

Among the 27,960 who were suspected of having VA – type cervical spondylosis and underwent VA color Doppler ultrasonography examination, 18 cases (0.06%) were found to have a duplicated VA origin. These patients were consisted of 5 males and 13 females, with an average age of 66.67 ± 12.24 years (range: 39–86 years). Most of the 18 patients included in the study complained headache or dizziness. With regard to cerebrovascular risk factors, 8 of 18 had hypertension, 5 had coronary heart disease, 3 had diabetes, 14 had hyperlipidemia, 2 had hyperhomocysteinemia. In these cases, left-sided VA duplication was present in 5 cases, and right-sided VA duplication was present in 13 cases. (Table 1 presented the demographics of the patients, as well as origin, level of fusion, diameter and PSV in each segment of the 18 duplicated VA).

### VA origin and level of entry into the transverse foramen

In 5 cases of left-sided VA duplication, one patient had both duplicated branches originated from the left subclavian artery (LSCA), the other 4 patients had one of the duplicated VA originated from the aortic arch (AOA), and the other originated from the LSCA. Among the 13 cases of right-sided VA duplication, one had their both branches originated from the AOA, one had different origin from the right subclavian artery (RSCA) and the AOA respectively; in the other 11 cases, both

**Table 1. Characteristics of the 18 patients with duplication of the VA.**

| Number | Gender | Age | Chief complain for VA ultrasound examination | Side | Medial limb | | | | Lateral limb | | | | Common limb | |
|---|---|---|---|---|---|---|---|---|---|---|---|---|---|---|
| | | | | | Origin | Level of fusion | Diameter (mm) | PSV (cm/s) | Origin | Level of fusion | Diameter (mm) | PSV (cm/s) | Diameter (mm) | PSV (cm/s) |
| 1 | Female | 59 | Ischemic cerebral vascular disease | Left | AOA | C5 | 0.26 | 90.6 | LSCA | C6 | 0.18 | 38.8 | 0.28 | 60.9 |
| 2 | Female | 72 | VA compression syndrome | Left | AOA | C5 | 0.21 | 77.6 | AOA | C6 | 0.24 | 56.8 | 0.28 | 54.1 |
| 3 | Male | 65 | Ischemic cerebral vascular disease | Left | AOA | C5 | 0.32 | 83.2 | LSCA | C6 | 0.27 | 50.7 | 0.35 | 79.2 |
| 4 | Male | 67 | VA compression syndrome | Left | AOA | C3 | 0.25 | 45.3 | LSCA | C6 | 0.23 | 75.5 | 0.28 | 58.2 |
| 5 | Female | 81 | VA compression syndrome | Left | AOA | C4 | 0.27 | 90.6 | LSCA | C6 | 0.26 | 51.8 | 0.35 | 49.6 |
| 6 | Male | 83 | Ischemic cerebral vascular disease | Right | AOA | C4 | 0.29 | N | RSCA | C6 | 0.23 | N | 0.36 | 56.8 |
| 7 | Female | 73 | VA compression syndrome | Right | RSCA | C5 | 0.33 | 64.9 | RSCA | C6 | 0.31 | 49.6 | 0.44 | 46.7 |
| 8 | Female | 56 | VA compression syndrome | Right | RSCA | C4 | 0.33 | 77.1 | RSCA | C6 | 0.26 | 40.6 | 0.31 | 50.7 |
| 9 | Female | 51 | VA compression syndrome | Right | RSCA | C4 | 0.15 | 46.4 | RSCA | C6 | 0.19 | 33.4 | 0.23 | 37.7 |
| 10 | Male | 64 | VA compression syndrome | Right | RSCA | C4 | 0.22 | 41 | RSCA | C6 | 0.24 | 44.6 | 0.35 | 44.6 |
| 11 | Male | 39 | Ischemic cerebral vascular disease | Right | RSCA | C4 | 0.23 | 42.6 | RSCA | C6 | 0.22 | 39.9 | 0.35 | 40.6 |
| 12 | Female | 67 | Ischemic cerebral vascular disease | Right | RSCA | C6 | 0.27 | 51.8 | RSCA | C4 | 0.27 | 39.9 | 0.34 | 44.60 |
| 13 | Female | 80 | VA compression syndrome | Right | RSCA | C5 | 0.27 | 113.2 | RSCA | C6 | 0.23 | 114.30 | 0.30 | 48.00 |
| 14 | Male | 52 | VA compression syndrome | Right | RSCA | C4 | 0.29 | 43.10 | RSCA | C6 | 0.22 | 33 | 0.38 | 42.60 |
| 15 | Female | 60 | VA compression syndrome | Right | RSCA | C4 | 0.2 | 58.2 | RSCA | C6 | 0.2 | 46.4 | 0.32 | 53.9 |
| 16 | Female | 44 | Ischemic cerebral vascular disease | Right | RSCA | C4 | 0.19 | 43.9 | RSCA | C6 | 0.2 | 61.5 | 0.29 | 49.4 |
| 17 | Female | 63 | Ischemic cerebral vascular disease | Right | RSCA | C4 | 0.18 | 51.8 | RSCA | C6 | 0.22 | 58.2 | 0.28 | 51.8 |
| 18 | Female | 70 | VA compression syndrome | Right | AOA | C4 | 0.26 | 50.7 | AOA | C6 | 0.26 | 41 | 0.4 | 49.60 |

Abbreviations: VA: vertebral artery; RSCA: Right subclavian artery; LSCA: left subclavian artery; AOA: aortic arch; PSV: peak systolic velocity

branches originated from the RSCA. In the 36 branches identified, 27 branches originated from the subclavian artery and the other 9 from AOA. The medial branch originated from the subclavian artery in 11 cases (61.1%), from the AOA in 7 cases (38.9%), from the right aortic arch in 5 cases, and from the left aortic arch in 2 cases. The lateral branches originated from the subclavian artery in 16 cases (88.9%), from the AOA in 2 cases (11.1%), and from both the left and right branches in 1 case. The proportion of double – stemmed arteries originating from the aortic arch was 9/38 (23.7%).

The VA typically passes through the transverse foramen of C6 and then ascends through the intervertebral foramen. If it does not enter C6, it is regarded as a variation. Therefore, the variation rate represents the proportion of VAs without C6 entry among the total number of VAs. The variation rate of the entry point of the medial branch (1 case at C6, 5 cases at

C5, 11 cases at C4, 1 case at C3) was 94.4%, while that of the lateral branch (17 cases at C6, 1 case at C4) was 5.6%. The convergence point of the two branches (5 cases at C5, 12 cases at C4, 1 case at C3) showed a 100% variation rate. The level of the contralateral VA entering the transverse foramen was at the C6 level in 16 cases and at the C5 level in 2 cases, with a variation rate of 11.1%.

### Diameter and PSV

The diameters of the medial and lateral branches were $0.25 \pm 0.05$ cm and $0.23 \pm 0.03$ cm respectively, the diameter of the convergent intervertebral segment was $0.32 \pm 0.05$ cm, and the diameter of the contralateral intervertebral segment was $0.33 \pm 0.05$ cm. Ultimately, no significant differences in diameter were observed when comparing the medial and lateral branches or the convergent and contralateral vessels ($P = 0.092$ and $0.548$ respectively). However, significant differences were observed when comparing the diameters of the medial and lateral branches with those of the contralateral vessels ($P < 0.01$). The PSV data of the medial and lateral branches were available in 17/18 patients. The mean PSV of the medial, lateral, and convergent branches was $63.06 \pm 21.71$ cm/s, $51.53 \pm 19.53$ cm/s, and $50.72 \pm 9.47$ cm/s respectively, while the mean PSV of the contralateral VA was $55.36 \pm 12.87$ cm/s. Significant differences were observed when comparing the PSV between medial and lateral branches ($P = 0.035$) and the medial and convergent branches ($P = 0.015$). The differences between the lateral branch and the convergent branch or between the convergent branch and the contralateral VA were insignificant ($P = 0.871$ and $0.129$ respectively).

Differences in vessel diameter and PSV parameters between patients with VA compression syndrome and ischemic cerebral vascular disease were compared, no significance were detected. These results could be found in S1 and S2 Tables.

### Discussion

The VA plays a crucial role in supplying blood to the head. Any alterations in its anatomical structure can have a significant impact on the blood supply of the posterior circulation. During neck surgeries, interventional diagnosis and treatment, it is of utmost importance to comprehensively understand the anatomical variations of this artery.

This knowledge is particularly relevant in the diagnosis of vertebrobasilar artery ischemia. In cases of VA duplication, there is greater variability in the course of the VA, like the level at which it enters the transverse foramen, and its internal diameter. Specifically, in such cases, the systolic flow velocity of the medial branch and the flow velocity after convergence are notably higher compared to those of the lateral branch [7].

Ultrasound, a non-invasive examination modality, offers several advantages, including repeatable, provides real-time imaging, and cost-effective. Thus, it serves as the preferred and routine examination for evaluating the VA. Ultrasound can clearly depict the course of the VA between the vertebrae and the influence of the vertebrae on it. It can also clearly visualize the intimal structure of the VA, vascular calcification, localized stenosis, and congenital malformations. Moreover, it can discern the blood flow direction of the VA, which is crucial for diagnosing subclavian artery steal syndrome. Based on the measured data, the blood flow volume per minute of the VA can be calculated to assess whether there is insufficient blood supply.

The embryogenesis of the VA commences on the 32nd day and is fully completed by the 40th day [8,9]. At the four-week-old embryonic stage, the vertebrobasilar artery has not yet been formed. Only two indistinct longitudinal neural arteries are located dorsal to the carotid arteries. These are formed through the anastomosis of seven intersegmental arteries corresponding to each vertebral body and the longitudinal plexus, existing solely in the form of a vascular plexus. Apparently, this cannot adequately satisfy the developmental demands of the posterior part of the brain. As development proceeds, the lateral seventh intersegmental artery dilates, thereby giving rise to the proximal subclavian artery and the origin of the VA. The dorsal branch of the seventh intersegmental artery develops into the V – 1 segment of the VA, and the plexiform anastomotic channels between the C6 - C1 vertebral bodies form the V – 2 segment of the VA. When

these embryonic intersegmental arteries undergo atypical regression or persistence, it leads to abnormal vascular origins [10–12]. In some reports, the development of duplicated VA is thought to originate from abnormal embryonic aortic arch development and the persistence of intersegmental arteries [8].

Unilateral VA duplication is estimated to occur in 0.79% − 0.295% of the population. The duplicated trunk on the left side typically originates from the aortic arch and the left subclavian artery, while the duplicated trunk on the right side usually derives from the right subclavian artery, although it can also branch from the right CCA [1,2]. Although unilateral VA duplication is the most prevalent, bilateral cases have also been reported [12,13]. In this study, the incidence of duplicated VA was only 0.06%, which is significantly lower than previously reported. This may be ascribed to the relatively larger number of VA cases detected by ultrasound compared with the past.

All cases in this study demonstrated unilateral origin of duplicated VA, with no occurrences of bilateral VA duplication. Left – sided VA duplication (80%) was more common than right – sided duplication. The aortic arch was the most frequent origin site in left – sided cases, while the right subclavian artery was the most common origin site in right – sided cases. No cases with origin from the CCA were identified. In cases of duplicated VA, the convergence point of the two main branches is generally at the C4 - C5 level [1,14]. In this study, the majority of the medial branches did not enter at the C6 level, and the variation rate of the entry level into the transverse foramen was as high as 94.4%. The variation rate of the entry point of the lateral branch was only 5.6%, and the variation rate of the convergence point of the two branches was 100%, with most located at the C3, C4, and C5 levels. The levels of contralateral VA entry into the transverse foramen were C6 in 16 cases and C5 in 2 cases, with a variation rate of 11.1%, which is consistent with previous reports [15].

Identifying the vascular origin and course is of particular importance for clinicians performing interventional procedures. An accurate assessment of the origin and course of the VA can effectively prevent surgical injuries and related complications.

Since duplicated VA is most commonly encountered in case reports or literature reviews [3], and based on the CT angiography measurements of 10 reported patients [1], the internal diameter or hemodynamic characteristics of the vessels in affected cases still await systematic exploration. When comparing the two main branches in cases of duplicated VA, the internal diameter of the medial branch generally tends to be larger than that of the lateral branch [2,16]. However, in this study, there was no significant difference in the diameter between the two branches. Instead, the diameters of the medial and lateral branches were significantly smaller than the internal diameter of the convergent VA, which may be attributed to the convergence of these two vessels.

When comparing the medial and lateral branches on the side of the duplicated VA with the confluent branch, significant differences in diameter were observed. This finding is of clinical significance, as smaller VAs are at an elevated risk of atherosclerosis, arterial stiffness, vascular weakening, lumen narrowing, and subsequent exacerbation of vertebrobasilar insufficiency [17]. In the event of severe stenosis or acute occlusion of the basilar artery, clinical intervention is required, and either the non-duplicated VA [16] or the larger branch of the duplicated VA should be selected as the interventional pathway. In terms of hemodynamics, the PSV of the medial branch is greater than that of the lateral branch and the confluent branch. The PSV of the medial branch of the left-sided duplicated VA is higher than that of the subclavian artery, which might contribute to ensuring an adequate blood supply. However, the pressure within the AOA may play a central role in VA rupture or aneurysm dissection [18]. This study also demonstrated that the PSV of the medial branch is greater than that of the lateral branch. The internal diameter of the medial branch is wider than that of the lateral branch, which could account for this finding.

Moreover, during the upward progression of the medial branch, these structures usually traverse the anterior edge of the transverse process and are located within the triangular area formed by the anterior scalene muscle, the longus colli muscle, and the subclavian artery, presenting relevant symptoms. Rotation or lateral flexion of the neck can force the VA to shift, resulting in alterations in traction, folding, and angulation, thereby narrowing the lumen, and simultaneously

stimulating or compressing the VA. Such actions can stimulate the sympathetic nerves on the VA wall, leading to VA spasm.

The VA duplication malformation is frequently asymptomatic. In this study, none of the patients exhibited occlusion, arterial stenosis, or dissection of the VA with dual origin. However, there have been reported cases associated with vascular dissection [19]. For instance, there was a report of a 51-year-old male who experienced an acute cerebellar vermis infarction due to medial branch dissection of the right-sided duplicated VA following minor neck trauma. Some reports also suggest that symptomatic thrombosis can occur on the left side in the duplicated VA branches [20]. Despite the high risk of cardiac embolism, symptoms improved with treatment. Compared to other forms of VA thrombosis or dissection, VA thrombosis or dissection in cases of VA duplication may lead to milder symptoms due to the compensatory effect of the remaining VA branches [21]. In cases of severe cervical spine injury, the VA is prone to rapid subluxation, deceleration, fracture of the cervical transverse foramen, or cervical flexion. When a motor vehicle accident is involved, the additional attachment points of the VA render individuals with VA duplication more susceptible to severe consequences [22].

This study is subject to several limitations. Firstly, these analyses are based solely on the outcomes of color Doppler ultrasonography, which might be somewhat subjective in nature. Additionally, the sample size of this study is relatively small, thereby emphasizing the necessity for more data. Also, this study did not provide correlation analysis between PSV and other cerebral perfusion parameters. Nonetheless, the data provide a valuable reference for clinically identifying hemodynamic abnormalities within this vascular variant and lay the groundwork for future research into its connection with posterior circulation ischemia risk. Nevertheless, these results provide novel insights into the alterations in the hemodynamic and physical characteristics of cases involving duplicated vertebral arteries (VA), and offer assistance to physicians conducting head and neck surgeries or cerebrovascular interventional procedures. Moreover, they also provide general perspectives on the diseases and anomalies of the vertebrobasilar system.

## Conclusion

In conclusion, although cases of duplicated VA are often asymptomatic, they can lead to hemodynamic aberrations, thereby increasing the risk of posterior circulation ischemia, triggering clinical symptoms, and augmenting patients' susceptibility to cerebrovascular disorders. When conducting invasive surgical procedures in the vicinity of the VA origin, such as cervical stenting, interventional procedures, or anterior cervical spine surgery, the diagnosis of duplicated VA is of considerable clinical significance. Such a diagnosis can guide appropriate treatment planning while enabling the avoidance of iatrogenic injury and blood loss. Color Doppler ultrasonography is a safe, non-invasive, and repeatable examination modality that lacks any contraindications. Moreover, it can measure hemodynamic parameters at a low cost. Consequently, it holds high diagnostic value for cases of duplicated VA.

## Supporting information

**S1 Table. Descriptive statistics of 18 subjects dichotomized into vertebral artery compression syndrome group and ischemic cerebral vascular disease group.**
(DOCX)

**S2 Table. Independent-samples T test of age and ultrasound measurement parameters between vertebral artery compression syndrome group and ischemic cerebral vascular disease group.**
(DOCX)

## Acknowledgments

The authors acknowledge all the patients for their participation in this study.

## Author contributions

**Conceptualization:** Li Liu.

**Data curation:** Xue Han, Wenbo Duan, Wanling Wen, Tian Lin.

**Formal analysis:** Li Liu.

**Investigation:** Xue Han, Wenbo Duan, Li Liu.

**Methodology:** Xue Han, Li Liu.

**Project administration:** Juan Du.

**Resources:** Dongxu Lu, Juan Du.

**Software:** Xue Han, Wanling Wen.

**Supervision:** Li Liu.

**Validation:** Xue Han, Wenbo Duan, Tian Lin, Juan Du.

**Writing – original draft:** Xue Han.

**Writing – review & editing:** Wenbo Duan, Wanling Wen, Tian Lin, Dongxu Lu, Juan Du, Li Liu.

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
