## [Decision Letter · Decision Letter 0]

3 Sep 2025

Dear Dr.  Liu,

Thank you for submitting your manuscript to PLOS ONE. After careful consideration, we feel that it has merit but does not fully meet PLOS ONE’s publication criteria as it currently stands. Therefore, we invite you to submit a revised version of the manuscript that addresses the points raised during the review process.

We look forward to receiving your revised manuscript.

Kind regards,

Priti Chaudhary, M.S.

Academic Editor

PLOS ONE

Journal Requirements:

“This study was supported by the Subject Assistance Program of the Strategic Support Force Medical Center (NO.21XK0101), whose receiver did not take part in this study.”

Additional Editor Comments:

**Authors are required to reply all the queries, raised by Reviewer 2. **
**Methodology is not clear. 1. How the patients were selected. **

**Comments to the Author**

1. Is the manuscript technically sound, and do the data support the conclusions?

Reviewer #1: Yes

Reviewer #2: Yes

2. Has the statistical analysis been performed appropriately and rigorously?

Reviewer #1: Yes

Reviewer #2: I Don't Know

3. Have the authors made all data underlying the findings in their manuscript fully available?

Reviewer #1: Yes

Reviewer #2: No

4. Is the manuscript presented in an intelligible fashion and written in standard English?

Reviewer #1: Yes

Reviewer #2: Yes

Reviewer #1: The manuscript was interesting and good written and discussed.

Introduction was good written.

Materials and method were good designed.

Results were good described and illustrated

Discussion was good written.

Reviewer #2: 1) the patient selection criteria are unclear to me. I am not aware of a disease that is named VA-cervical spondylosis. Please make it clear to the reader what is the indication for these patients to have underwent the ultrasound examination? Is it VA compression syndrome symptoms? If so, clarify.

2) The clinical relevance of duplicate VA is not at all explored further in the sample set, while the manuscipt would be substantially improved in valor if the authors would substantiate the claim in their dataset that these duplications are associated with other vascular malformations?

3) The clinical relevance of just a comparison of the two limbs in 1 patient compared to the other VA is in my opinion unclear if the readers has no idea of the other parameters that are relevant to cerebral perfusion.

4) The methodology of measurement of PSV in the vertebral artery may be dependent of the insonation angle. There is no reference of the way that this was controlled for between the duplicate arteries. How sure can we be that differences in flow are not based on technical flaws? Please clarify.

**Do you want your identity to be public for this peer review?** For information about this choice, including consent withdrawal, please see our Privacy Policy

Reviewer #1: No

Reviewer #2: No

---

## [Author Response · Author response to Decision Letter 1]

17 Oct 2025

Additional Editor Comments:

Authors are required to reply all the queries, raised by Reviewer 2.

Methodology is not clear.

1. How the patients were selected. The study included patients who had undergone Color Doppler Ultrasonography of VA in suspicion of VA compression syndrome or ischemic cerebral vascular diseases.

2. How sample size calculation done. Kindly clarify.

First of all, this is a retrospective study, and the incidence of VA duplication was not quite clear, so that sample size was not rigidly calculated. This had been pointed out in limitation part.

Secondly, to get our study subjects, we screened all VA ultrasound examination records in our center during January 2019 to December 2022. And indications for VA ultrasound examination were patients with complains of dizziness or vertigo that in suspicion of VA compression syndrome and patients with evidence or risk of ischemic cerebral vascular disease including but not limited to acute ischemic stroke or transient ischemic attack. All patients we found had been included.

Comments to the Author

Review Comments to the Author

Reviewer #1: The manuscript was interesting and good written and discussed.

Introduction was good written.

Materials and method were good designed.

Results were good described and illustrated

Discussion was good written.

Reviewer #2:

1) the patient selection criteria are unclear to me. I am not aware of a disease that is named VA-cervical spondylosis. Please make it clear to the reader what is the indication for these patients to have underwent the ultrasound examination? Is it VA compression syndrome symptoms? If so, clarify.

Response to reviewer: The indications for VA ultrasound examination were patients with complains of dizziness or vertigo that in suspicion of VA compression syndrome and patients with risks or evidence of ischemic cerebral vascular disease including acute ischemic stroke or transient ischemic attack.

2) The clinical relevance of duplicate VA is not at all explored further in the sample set, while the manuscript would be substantially improved in valor if the authors would substantiate the claim in their dataset that these duplications are associated with other vascular malformations?

Response to reviewer: Thank you for the suggestion. We totally agree with you on the importance of investigation about the relevance between VA duplication and other vascular malformations, however, within our limited sample size, we found only 4 of 18 subjects underwent head and neck CTA examination and 2 other patients had brain MRA examination, all of them reported no other vascular malformations. And we have put this in the limitation part. Further study with prospective design might offer a better answer to this question.

3) The clinical relevance of just a comparison of the two limbs in 1 patient compared to the other VA is in my opinion unclear if the readers has no idea of the other parameters that are relevant to cerebral perfusion.

Response to reviewer: We agree that direct correlation with cerebral perfusion parameters would provide a more comprehensive picture. PSV is the cornerstone parameter for diagnosing arterial stenosis in Doppler ultrasonography. The main hypothesis of our study was that the two stems of duplicated vertebral artery, which potentially having different embryological origins, may exhibit inherent hemodynamic differences. So the primary clinical significance of our data lies in reporting the hemodynamic features of this specific anatomical variant. By comparing the PSV of each stem to that of a normal, single VA, we aim to answer a critical clinical question: What constitutes a 'normal' versus 'pathologically elevated' PSV in this variant? This is directly relevant to the sonographer or vascular neurologist in daily practice. Without such reference data, there is a risk of either over-diagnosing stenosis in a high-flow but normal stem, or under-diagnosing a true stenosis. Identifying a stem with significantly elevated PSV can signal the need for closer monitoring or further investigation with cross-sectional imaging (e.g., CTA or MRA) to rule out clinically significant stenosis that could impact posterior circulation perfusion.

We also added the following paragraph into the limitation part:

“One limitation of this study is the lack of correlation between our PSV measurements and cerebral perfusion parameters.”

4) The methodology of measurement of PSV in the vertebral artery may be dependent of the insonation angle. There is no reference of the way that this was controlled for between the duplicate arteries. How sure can we be that differences in flow are not based on technical flaws? Please clarify.

Response to reviewer: During the PSV measurement procedure, standardized device parameters were ensured and preset optimizations were applied. A unified preset condition for the vertebral artery was used. The Doppler gain was adjusted to clearly display the spectral waveform outline while avoiding background “snowy” noise. Insufficient gain may result in incomplete spectral filling, whereas excessive gain introduces noise. The wall filter was adjusted to retain true low-velocity flow signals while adequately filtering low-frequency signals caused by vessel wall motion. Excessive filtering may truncate diastolic flow and lead to measurement errors. Doppler angle correction was consistently performed, with the correction line aligned parallel to the vessel wall. For the V2 segment of the vertebral artery, the straight portion between transverse foramina was generally used as a reference. The Doppler angle θ was maintained at ≤ 60°. The sample volume size was set to cover most of the vessel lumen without extending beyond the vessel wall (typically 1.5–2.0 mm). It was positioned in the center of the vessel, away from near-wall regions, to obtain core flow velocities representative of laminar flow. This content has been included in the Methods section.

---

## [Editor Report · Decision Letter 1]

23 Oct 2025

Anatomic and Hemodynamic Characterization of Vertebral Artery Duplication via Color Doppler Ultrasonography

PONE-D-25-15044R1

Dear Dr. Li Liu,

We’re pleased to inform you that your manuscript has been judged scientifically suitable for publication and will be formally accepted for publication once it meets all outstanding technical requirements.

Kind regards,

Priti Chaudhary, M.S.

Academic Editor

PLOS ONE
---

## [Editor Report · Acceptance letter]

PONE-D-25-15044R1

PLOS ONE

Dear Dr. Liu,

I'm pleased to inform you that your manuscript has been deemed suitable for publication in PLOS ONE. Congratulations! Your manuscript is now being handed over to our production team.

Kind regards,

on behalf of

Dr. Priti Chaudhary

Academic Editor

PLOS ONE